# Molecular analysis of the ribosome recycling factor ABCE1 bound to the 30S post-splitting complex

Elina Nürenberg-Goloub[1,†], Hanna Kratzat[2,†], Holger Heinemann[1,†], André Heuer[2], Peter Kötter[3], Otto Berninghausen[2], Thomas Becker[2], Robert Tampé[1,*] & Roland Beckmann[2,**]

## Abstract

Ribosome recycling by the twin-ATPase ABCE1 is a key regulatory process in mRNA translation and surveillance and in ribosome-associated protein quality control in Eukarya and Archaea. Here, we captured the archaeal 30S ribosome post-splitting complex at 2.8 Å resolution by cryo-electron microscopy. The structure reveals the dynamic behavior of structural motifs unique to ABCE1, which ultimately leads to ribosome splitting. More specifically, we provide molecular details on how conformational rearrangements of the iron–sulfur cluster domain and hinge regions of ABCE1 are linked to closure of its nucleotide-binding sites. The combination of mutational and functional analyses uncovers an intricate allosteric network between the ribosome, regulatory domains of ABCE1, and its two structurally and functionally asymmetric ATP-binding sites. Based on these data, we propose a refined model of how signals from the ribosome are integrated into the ATPase cycle of ABCE1 to orchestrate ribosome recycling.

**Keywords** ABC proteins, ribosome recycling; molecular machines; mRNA surveillance; ribosome-associated quality control

**Subject Categories** Structural Biology; Translation & Protein Quality

**The EMBO Journal (2020) 39: e103788**

## Introduction

Protein biosynthesis via mRNA translation is a fundamental process in living cells. Strikingly, translation is interlaced in a complex network of cellular pathways including mRNA surveillance, ribosome-associated quality control, and ribosome biogenesis (Bassler & Hurt, 2019; Joazeiro, 2019; Nürenberg-Goloub & Tampé, 2019). These crucial pathways maintain protein, mRNA, and ribosome homeostasis (Young *et al*, 2015; Mills *et al*, 2016), induce organelle turnover (Wu *et al*, 2018), assist embryonic development (Coelho *et al*, 2005; Chen *et al*, 2006), and are also linked to various

diseases including ribosomopathies and cancer (Tahmasebi *et al*, 2018; Aspesi & Ellis, 2019). Accordingly, each of the four phases of translation—initiation, elongation, termination, and ribosome recycling—as well as the transitions between them must be under rigorous control. While the first three phases are directly involved in protein biosynthesis and have therefore been extensively studied, ribosome recycling has only recently been structurally and functionally characterized (Hellen, 2018). Herein, the conserved and essential ATP-binding cassette (ABC)-type twin-ATPase ABCE1 plays the key role for Archaea and Eukarya (Pisarev *et al*, 2010; Barthelme *et al*, 2011; Shoemaker & Green, 2011). ABCE1 recycles canonical 70S/80S post-termination complexes (post-TCs) after stop codon-dependent termination and non-canonical post-TCs during mRNA surveillance and resumption of translation after cellular stress. In both cases, a decoding A-site factor (archaeal/eukaryotic release factor 1 (a/eRF1) or its homologue a/ePelota, respectively) is delivered to the ribosomal A-site by a translational GTPase (aEF1/eRF3 or aEF1/Hbs1, respectively) and forms an interaction platform for ABCE1 to establish the 70S/80S pre-splitting complex (pre-SC; Becker *et al*, 2012; Preis *et al*, 2014; Brown *et al*, 2015; Shao *et al*, 2016). In concert with the A-site factor, ABCE1 splits the pre-SC into the small (SSU) and large (LSU) ribosomal subunit. In Eukarya, other components of the post-TC stay associated with the ribosomal subunits and are subsequently recycled by additional factors (Pisarev *et al*, 2010; Skabkin *et al*, 2010). Canonical termination, which includes peptide release by eRF1, yields 40S-mRNA-deacylated tRNA complexes and free 60S subunits whereas ribosome recycling of non-canonical post-TCs in the presence of Pelota results in 40S-mRNA and 60S-peptidyl-tRNA complexes due to Pelota's incapacity to release peptides. Moreover, Pelota/Hbs1/ABCE1 not only acts in the splitting of stalled (Shoemaker & Green, 2011), but also vacant (van den Elzen *et al*, 2014), and newly synthesized ribosomes (Strunk *et al*, 2012). Immediately after splitting, an ABCE1-bound 30S/40S post-splitting complex is formed (Kiosze-Becker *et al*, 2016; Heuer *et al*, 2017), in which ABCE1 may remain for a defined time span (Nürenberg-Goloub *et al*, 2018; Gouridis *et al*, 2019) to prevent re-association of the LSU (Heuer *et al*, 2017). Additionally, ABCE1 has been shown to interact with initiation factors and is

1   Institute of Biochemistry, Biocenter, Goethe University Frankfurt, Frankfurt a.M., Germany
2   Department of Biochemistry, Gene Center, Ludwig-Maximilians University Munich, München, Germany
3   Institute for Molecular Biosciences, Biocenter, Goethe University Frankfurt, Frankfurt a.M., Germany
    *Corresponding author. Tel: +49 069 798 29475; E-mail: tampe@em.uni-frankfurt.de
    **Corresponding author. Tel: +49 089 218 076900; E-mail: beckmann@genzentrum.lmu.de
    †These authors contributed equally to this work

assumed to promote their recruitment to the SSU (Dong *et al*, 2004; Chen *et al*, 2006), thus linking ribosome recycling to translation initiation.

A key question is which molecular mechanism is employed by ABCE1 as an ABC-type ATPase. All members of the ABC superfamily utilize the energy of ATP binding and hydrolysis generated in two conserved nucleotide-binding sites (NBS) and are ubiquitously found in numerous cellular processes. These include transport of a limitless range of substrates across membranes, chromatin remodeling, DNA repair, or modulation of ribosomal complexes. The NBSs are formed at the interface of two nucleotide-binding domains (NBDs), which are arranged reciprocally (Hopfner, 2016). ABCE1 additionally possesses an essential N-terminal iron–sulfur cluster domain (FeSD) (Barthelme *et al*, 2007) and a composite hinge region, which comprises a hinge 1 stretch between the NBDs and a hinge 2 stretch at the C terminus, and connects the two NBDs. A unique helix-loop-helix (HLH) insertion in NBD1 distinguishes it from the otherwise superimposable NBD2 (Karcher *et al*, 2008). The two functionally asymmetric NBSs have distinct roles during ribosome recycling (Nürenberg-Goloub *et al*, 2018) and can adopt multiple isoenergetic conformational states (Gouridis *et al*, 2019). We speculated that the state of the ribosome and the dynamic transitions during ribosome recycling (from pre-splitting to post-splitting states) can be precisely sensed by ABCE1 and are coupled to rearrangements in the NBSs (Nürenberg-Goloub *et al*, 2018).

To gain molecular information about the post-splitting complex, we solved the structure of the archaeal post-SC by cryogenic electron microscopy (cryo-EM) to an overall resolution of 2.8 Å. Our structure of ABCE1 bound to the 30S small ribosomal subunit allowed a thorough analysis of this asymmetric ABC protein in the nucleotide-occluded conformation at the level of individual residues. The NBSs of ABCE1 adopt the closed, nucleotide-occluded state with two ATP-mimicking $Mg^{2+}$-AMP-PNP molecules bound in both NBSs. In general, both catalytic sites superimpose well with marginal deviations. Comparison with the best-resolved structure of the pre-SC (Brown *et al*, 2015) reveals that the functionally important hinge region opens up in the post-SC, allowing ABCE1 to adopt the nucleotide-occluded state. Our high-resolution cryo-EM structure explains how this conformational change can induce an allosteric crosstalk from the SSU into the two functionally distinct NBSs, giving new insights into how the different stages of ribosome recycling are linked to ABCE1's ATPase cycle.

## Results

### Assembly of the post-splitting complex

To obtain archaeal post-SCs, we actively split isolated native *Thermococcus celer* (*T. celer*) 70S ribosomes using recombinant ABCE1, aRF1, and aPelota from the related archaeon *Saccharolobus solfataricus* (*S.s.*), thus ensuring to resemble the cellular recycling route for all ribosomes present in the native mixture: ribosomes with the A-site occupied by a stop codon (aRF1), a sense codon (e.g., in stalled ribosomes) or vacant ribosomes (aPelota). Thereby, we circumvented a low-$Mg^{2+}$ and high $K^+$ treatment necessary for facilitated ribosome splitting as previously performed in yeast (Heuer *et al*,

2017). To stabilize the post-SC, a well-characterized, hydrolysis-deficient ABCE1 mutant was used. This mutant, with both catalytic glutamates being substituted by alanine (E238A/E485A, short IIEA), efficiently split 70S ribosomes and remained quantitatively bound to 30S subunits (Nürenberg-Goloub *et al*, 2018) (Fig 1A). Notably, 70S from *S. solfataricus* are intrinsically instable (Barthelme *et al*, 2011) and thus unsuitable for our *in vitro* splitting approach.

The purified 30S-ABCE1$^{IIEA}$ post-SC was subjected to single-particle cryo-EM analysis. 3D classification revealed that the vast majority (97%) of 30S particles were associated with ABCE1$^{IIEA}$. This class was refined to an average resolution of 2.8 Å (Fig 1B). Local resolution assessment showed that the body of the 30S formed a very rigid structure whereas the 30S head and ABCE1 showed flexibility and lower resolution (4–6 Å) (Fig EV1). However, using focused refinement, the local resolution was improved to 3.0 Å for ABCE1 and to 2.8 Å for the 30S head. This allowed to build a complete molecular model for the *T. celer* SSU associated with ABCE1 (Figs 1C and EV1).

### Molecular model of the *Thermococcus celer* small ribosomal subunit

The *T. celer* 30S ribosome structure comprises 1,485 nucleic acid residues of 16S ribosomal RNA (rRNA) (Appendix Fig S1) and 28 ribosomal proteins (Fig EV2A). As an initial template, we used the structure of the closely related *Pyrococcus furiosus* (*P.fu.*) ribosome modeled at 6.6 Å resolution (Armache *et al*, 2013), to which *T. celer* rRNA shows 96% and ribosomal proteins 78–95% sequence identity, respectively. All residues were manually exchanged to the correct *T. celer* sequence and fitted into the electron density map. Several protein N and C termini as well as loop regions were built *de novo*. This was possible for the entire 30S subunit except for rRNA and proteins forming the beak (eL8, eS31, and parts of h33), which is known to be the most flexible moiety of the SSU (Fig EV1).

Interestingly, we discovered a previously unobserved density for a ribosomal protein on the 30S platform, which was identified as a so far uncharacterized protein and its structure was built *de novo* (Figs 1B and EV2). The 59 amino acid (aa) long protein (6.6 kDa) is located in a cleft between uS2, uS5, and uS8, close to helix (h) 36 and h26/h26a of 16S rRNA. There, it occupies the same position as eS21 in the *Saccharomyces cerevisiae* (*S.c.*) 40S ribosome, whereas in the 30S ribosome from *Escherichia coli* (*E.c.*), the equivalent position is not covered (Fig EV2B). The sequence matches UniProtKB: A0A218P055 (A0A218P055_THECE) and contains a zinc-binding zinc ribbon domain, for which we could assign density for two bound zinc ions. It is conserved in other archaeal species, yet sequence identity with eS21 is rather low (Fig EV2C) with 7% for the full-length protein, but 27% for residues 10–24 representing the zinc ribbon. In accordance with the universal nomenclature for ribosomal proteins (Ban *et al*, 2014), we will refer to the identified protein as eS21.

### The architecture of the post-splitting complex is conserved among Eukarya and Archaea

Binding to 70S/80S ribosomes in pre-splitting and to 30S/40S ribosomes in post-splitting complexes is already known to be mainly mediated by the ABCE1-specific HLH motif and hinge region

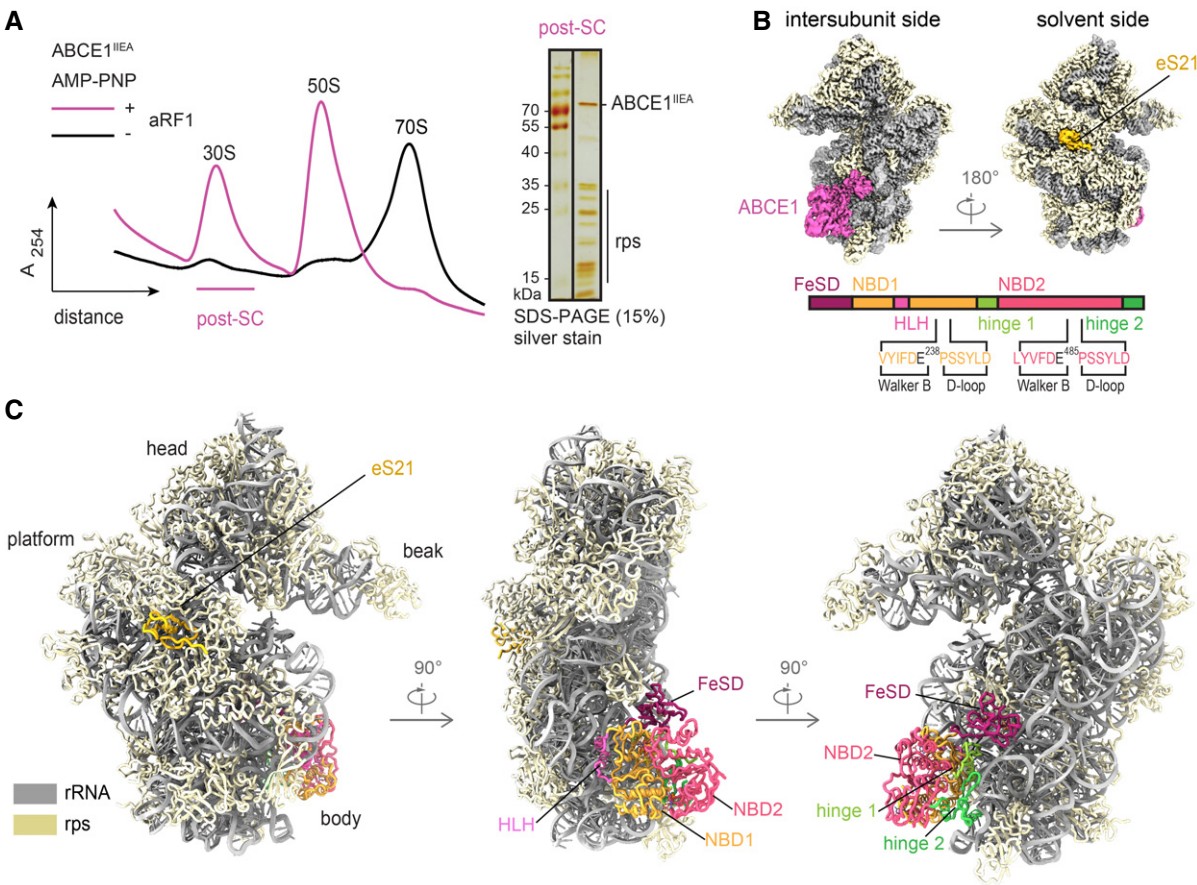

**Figure 1. *In vitro* assembly and cryo-EM structure of the archaeal post-splitting complex.**

A  ABCE1$^{IIEA}$ efficiently splits 70S ribosomes in the presence of AMP-PNP and aRF1/aPelota. The 30S population contains a stoichiometric ratio of ABCE1 and ribosomal proteins, forming the post-splitting complex. rps: small subunit ribosomal proteins.

B  Cryo-EM density of the post-SC highlights the archaeal ribosomal protein eS21 and ABCE1. Domain architecture of ABCE1 including the mutation sites is shown below.

C  Molecular model of the archaeal post-SC, domain colors as in (B).

Data information: In (A), the gradient profiles are representative for the respective nucleotide condition.

contacting the body of the SSU. Upon transition from the pre- to the post-splitting state, the NBSs move from a semi-open to a fully closed, nucleotide-occluded state. Concomitantly, the FeSD rotates around a cantilever toward the decoding site of the SSU close to rRNA helix h44 (Heuer *et al*, 2017).

The overall architecture of the archaeal post-SC is similar to the yeast 40S-ABCE1 complex (Heuer *et al*, 2017) showing the same hallmarks. The FeSD occupies a position close to rRNA h44, hinge region and HLH motif anchor the NBDs to the 30S body, and the two NBSs are in a closed conformation. Yet, the resolution of the archaeal post-SC (2.8 Å overall) is significantly higher than the one of the yeast post-SC (3.9 Å overall), especially in NBSII and the hinge region, thus allowing to describe interactions between ABCE1 and the SSU as well as interactions between the two NBSs on a molecular level. These molecular insights allowed us to draw conclusions and make predictions about the allosteric crosstalk between the two NBSs of ABCE1 as well as ABCE1 and the ribosome. Moreover, these insights guided the corresponding functional studies (see below).

## The FeSD domain establishes inter- and intramolecular interactions specific for the post-SC

Based on the high-resolution data, we can delineate crucial interactions between the FeSD domain, NBD1, hinge 1, and the 30S ribosomal subunit. The FeSD is embedded in a pocket between rRNA h44, the h5-h15 junction, and the universally conserved ribosomal protein uS12 (Fig 2A). The majority of FeSD interactions with the ribosome are conserved, while the loop regions of the FeSD opposite of the ribosome (e.g., L36-K43) are variable in sequence and structure, underlining the significance of the interaction of the FeSD with the ribosome (Fig EV3A, Appendix Fig S2). The majority of interactions are formed by salt bridges and hydrogen bonds established between conserved residues in ABCE1 (R2, K15, N17, E19, K59) and the phosphate backbone as well as 2'OH groups of rRNA (Fig 2A). Similarly, also the interaction sites between ABCE1 and uS12 are conserved (P25, R28, and S29 of ABCE1 to Q76 and H100 of uS12) (Fig 2A). Interestingly, we observed a few cases where the ribosome and ABCE1 co-evolved to maintain the interaction pattern. For

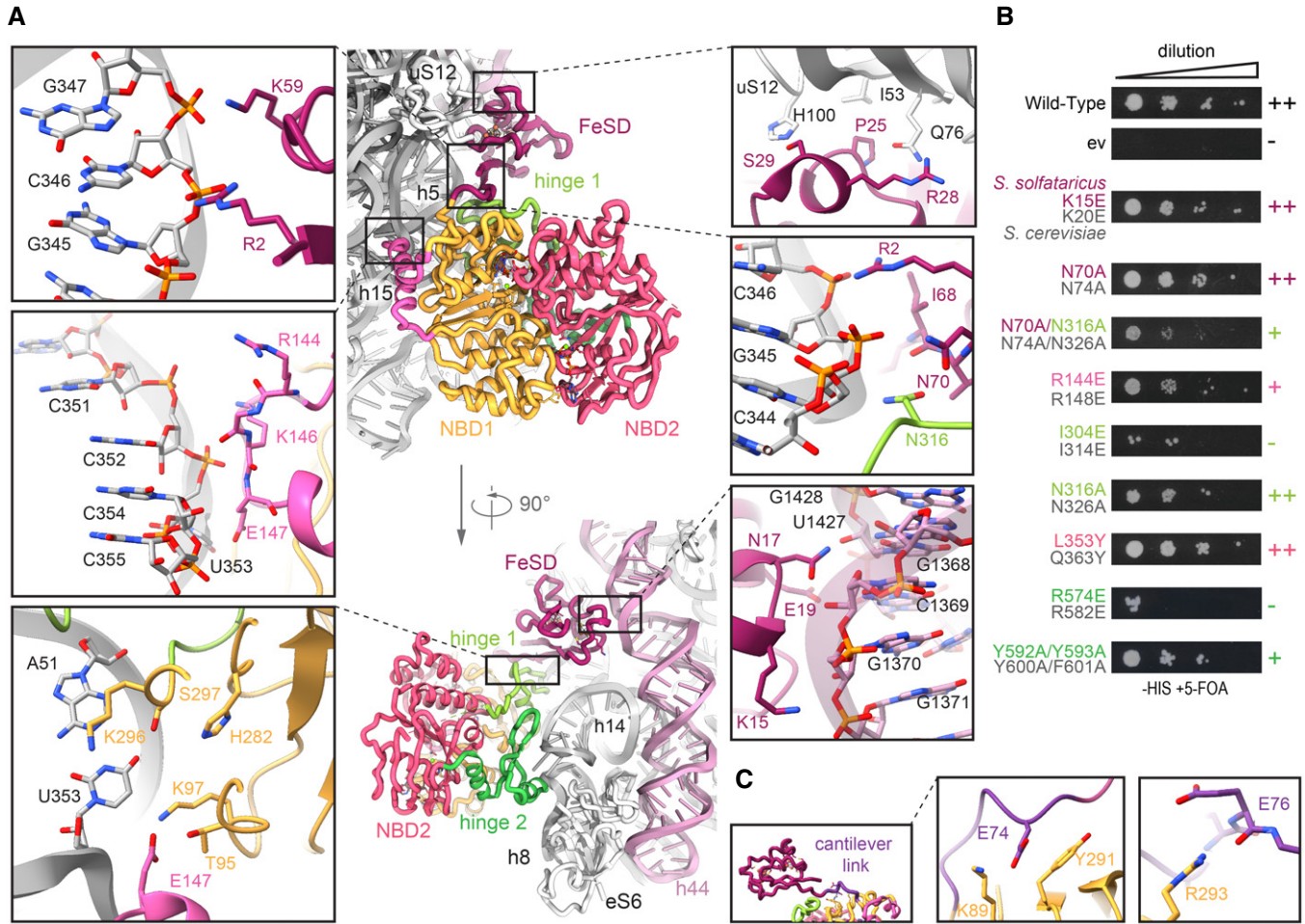

**Figure 2. The conserved ABCE1-30S interface is formed by essential interactions.**

A  Zoom-ins into ABCE1-30S connections. Most interactions are salt bridges or H-bonds between ABCE1 residues and the rRNA phosphate backbone. The FeSD cluster domain contacts rRNA h5 via R2 and K59, interacts with uS12 via S29 and R28, and contacts h44 by N17 and K15. The helix-loop-helix motif connects to rRNA h15 via R144 and E147. The positioning of the cantilever is stabilized by an interaction network of R2, I68, and N70 with N316 of hinge 1 and rRNA h5.

B  Yeast survival of ABCE1 variants (*S. solfataricus* colored, *S. cerevisiae* in gray). Most residues connecting to 30S in the post-SC show a growth defect when exchanged for a small one (alanine) or a negative charge (glutamate). ++ no effect, + growth defect, − lethal.

C  The cantilever link forms salt bridges of E74 and E76 with NBD1 residues K89 and R293, respectively.

Data Information: In (B), data are representative for a set of two independent experiments.

example, the interaction between S29 of ABCE1 and H100 of uS12 is substituted by the contact of K36 (ABCE1) with N99 (uS12) in yeast (Fig EV3B), underlining the importance of an interaction at this position for re-orientation of the FeSD after ribosome splitting.

The FeSD is linked to the main twin-ATPase body via a flexible linker connecting the cantilever β-sheet β4 with NBD1 (Fig 2C, Appendix Fig S2). This linker (D73-V79 in *S. solfataricus*) forms an α-helix in free ABCE1 and the pre-SC (Karcher *et al*, 2008; Brown *et al*, 2015), but unfolds into a loop in the post-SC. As in the yeast post-SC (Heuer *et al*, 2017), this cantilever helix is also unfolded in *S. solfataricus*. At high resolution, we deciphered a chain of inter- and intramolecular interactions that are a consequence of FeSD repositioning after splitting. We observed a similar stabilization of the cantilever loop by an interaction of Y291 in NBD1 (Y301 in *S.c.*) with the backbone of E74 (N78 in *S.c.*) (Fig 2C, Appendix Fig S2). In

our high-resolution structure, we identified additional stabilizing contacts for the cantilever loop. E74 also interacts with the side chain of K89 (NBD1) and the carbonyl group of E76 binds the guanidino group of R293 (NBD1) (Fig 2C). Moreover, an interaction network is formed between R2 (R7 in *S.c.*) at the N terminus, I68 and N70 (N74 in *S.c.*) of the cantilever β-sheet β4, and N316 (N326 in *S.c.*) in hinge 1, as well as the phosphate groups of G345 and G346 in rRNA h5 (Fig 2A). In yeast, the mutations Y301A and R7A impair the anti-association activity of ABCE1 *in vitro* and are synthetically lethal *in vivo* (Heuer *et al*, 2017). Additionally, we confirm synthetic lethality of N74A with N326A (Figs 2B and EV3C).

Taken together, closure of the NBSs displaces the FeSD, which leads to new interactions of the cantilever β-sheet and the cantilever loop with the ribosome, NBD1 and hinge 1. This allows for an allosteric communication of post-SC formation to the NBSs.

## Hinge 2 serves as a linchpin during ribosome splitting

The NBDs of ABCE1 are located at the body of the 30S subunit with main anchor points contributed by the HLH motif (to h15) and the dipartite hinge region (to junction of h8 and h14) (Fig 2A). In stark contrast to the pre-splitting complex, the HLH is displaced from its contact point at h5 by 16 Å toward h15. In the post-SC, h15 is in contact with the loop containing two basic residues (R144-G145-K146-E147) between helices α6 and α7 (Fig 2A). A charge reversal of the respective arginine in yeast (R148E) leads to a substantial growth defect, confirming this important position (Figs 2B and EV3C). The other residues in the HLH loop rather stabilize an interaction formed by NBD1 with U353, which flips out of h15 and forms a Watson-Crick base pair with A51 in h5, establishing the h5-h15 junction. Multiple residues (T95, K97, E147, H282, K296, and S297) are facing this base pair, suggesting that this specific tertiary structure is precisely monitored by NBD1 and the HLH motif of ABCE1 (Fig 2A). In contrast to yeast, no contacts are observed between ABCE1 and eS24, which is also present but significantly shorter at its C terminus in *T. celer*.

The ABCE1-specific hinge region is subdivided into hinge 1 (*S. solfataricus* 298–325) and hinge 2 (*S. solfataricus* 547–594; Appendix Fig S2). Interactions with the ribosome are mainly established by hinge 2. Hinge 1 connects NBD1 and NBD2 via a flexible linker (*S. solfataricus* 326–338), which is—as in other structures—only partially visible. Similar to the HLH/NBD1 region, hinge 2 also recognizes a special tertiary structure of the rRNA. It binds at the junction between rRNA helices h8 and h14, where A329 flips out of h14 and stacks upon the ribose of A138 in h8. The geometry is read out by the conserved R565 forming a cation-π-stack with A138 (Fig 3A and D, Appendix Fig S2). Notably, this interaction is maintained during ribosome splitting (Fig 4), and exchange of the corresponding residue (R573E) leads to loss of function in yeast (Karcher *et al*, 2008). Hence, the *S. solfataricus* ABCE1$^{R565E}$ mutant (Appendix Fig S3) was unable to bind 30S ribosomes (Figs 3E and EV4A) and failed to split 70S ribosomes (Figs 3F and EV4B), whereas the ATPase activity was similar to wild-type ABCE1 (Fig 3G).

The second main contact to the h8-h14 junction is formed by a salt bridge between R574 and the phosphate of U328 (Fig 3A and C). Moreover, R572 and N305 in hinge 1 stabilize the interaction network around this junction on the side of h14 (Fig 3C), while K577, S580, and R584 are in close contact to h8 (to G137 and A139) (Fig 3A and D). Further, hinge 2 forms an additional interaction site with eS6 by stacking Y581 against R69 (eS6) (Fig 3D). This interaction also occurs in yeast between Q589 and K58 (eS6), indicating a co-evolution of ABCE1 ribosome interactions as previously described for FeSD and uS12 (Fig EV3D).

While the hinge 2 region serves as a constant linchpin to the ribosome, the interaction pattern of hinge 1 is substantially altered compared to the pre-SC. In hinge 2, only R574 switches from U329 in the pre-SC to the adjacent U328 in the post-SC, while all other residues remain with their respective interaction partners (Fig 4A). In contrast, the entire hinge 1 region opens up relative to hinge 2, which results in a 5 Å shift of the hinge 2 β-sheets β25 and β26 (Fig 4A, Appendix Fig S2) and a 10 Å movement of hinge 1 helix α15. Together with the movement of the HLH (Fig 4B) and the FeSD, this conformational rearrangement, which we term "hinge opening", leads to the formation of new ribosomal contacts specific

for the post-SC. Thus, α15 of hinge 1 binds U328 and the conserved N316 binds to A314 as well as the phosphates of G343 and G345 close to the h5-h15 junction (Fig 2A). As mentioned above, U328 also contacts R574 in hinge 2 (Fig 3C) while N316 is connected to the rearranged cantilever loop of the FeSD. Consequently, the FeSD, hinge 1, and hinge 2 form a post-SC state-specific intricate interaction network.

Functional analyses and lethality screens confirm the essential role of the hinge 2 region for ABCE1 function. As mentioned before, ABCE1$^{S580E}$ (Appendix Fig S3) exhibits wild-type ATPase activity (Fig 3G) but neither binds to 30S ribosomes (Figs 3E and EV4A) nor splits 70S ribosomes (Figs 3F and EV4B). Additionally, the corresponding mutant is lethal in yeast (S588E) (Karcher *et al*, 2008). Interestingly, S580 is the N-terminal residue of helix α25 and does not directly interact with the ribosome but points toward α25 (Fig 3D). Thus, the mutation to glutamate at this position inhibits ribosome binding via destabilization of helix α25 rather than by direct repulsion. The importance of R574 for ribosome recognition is confirmed by our plasmid-rescue analysis in yeast, demonstrating that the respective R582E mutation is lethal (Figs 2B and EV3C).

## Structural asymmetry of the nucleotide-binding sites

Apparently, ABCE1 can act as timer for ribosome recycling (Heuer *et al*, 2017; Nürenberg-Goloub *et al*, 2018). During this process, the NBSs receive and integrate signals about the state of the ribosome, e.g., discriminate between pre-splitting and post-splitting complexes. In the post-SC, both NBSs have mainly been observed in the closed state (Gouridis *et al*, 2019), coinciding with a movement of the FeSD (Kiosze-Becker *et al*, 2016; Heuer *et al*, 2017) as initially suggested (Becker *et al*, 2012). Yet, in all obtained cryo-EM structures of pre- and post-SCs, the identity of the bound nucleotides, especially in NBSII, remained unclear. Based on our high-resolution data, we can resolve both catalytic pockets and unambiguously identify the non-hydrolysable ATP-analogue AMP-PNP complexed with a Mg$^{2+}$ ion in each NBS (Figs 3H–J and EV5). In agreement with the yeast post-SC and the structures of symmetric ABC-type NBD dimers (Lammens *et al*, 2011; Korkhov *et al*, 2012), AMP-PNP is sandwiched between the typical conserved motifs of ABC-type ATPases. In NBSI, the A-loop residue Y83 stacks on the purine base, which is contacted by the aliphatic part of D459 adjacent to the signature motif of the opposite NBD2. In addition, the ribose is stabilized by stacking with F88 (Fig 3I). The γ-phosphate is directly contacted by N108 (Walker A), H269 (His-switch), S461-G463 (signature motif), and Q167 (Q-loop), while T113 (Walker A) and D237 (Walker B) coordinate the Mg$^{2+}$ ion. Analogous residues are superimposable in NBSII, i.e., we find that N377 (Walker A), S214, G216 (signature motif), and H518 (His-switch) coordinate the γ-phosphate while Q411 (Q-loop), T382 (Walker A), and D484 (Walker B) contact the Mg$^{2+}$ ion (Fig 3J). Notably, the characteristic A-loop is degenerated in NBSII of most (but not all) organisms, featuring aliphatic or even polar residues (Gerovac & Tampé, 2019). Despite the degenerated A-loop (L353 instead of the aromatic residue), the accommodation of the purine base is similar to the one observed in NBSI (Fig 3H). The base is sandwiched between L353 and I212 adjacent to the signature motif of NBD1. Yet, we hypothesized that higher flexibility of the nucleotide in NBSII due to the degenerated A-loop might explain (i) the reduced intrinsic ATPase activity in

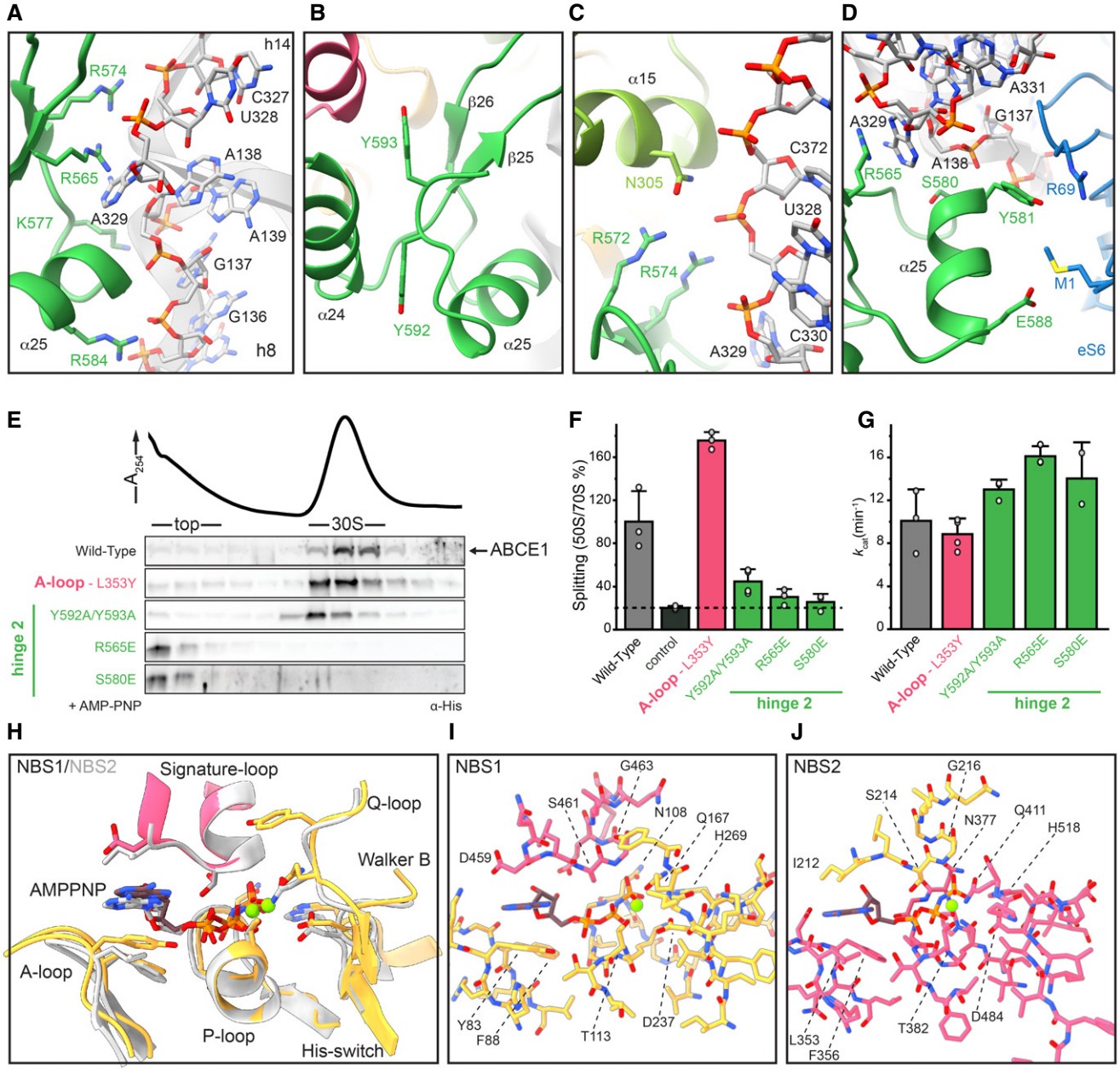

**Figure 3. Structural and functional analysis of the hinge regions and NBSs.**

A–D  Hinge 2 (emerald) residues interacting with the ribosome. R565E forms a conserved cation-π-stacking with A329 of h8; R574 forms a salt bridge with the phosphate backbone of U328 in h14. Aromatic C-terminal residues Y592 and Y593 adopt a parallel coordination. R572 of hinge 2 and N305 of hinge 1 (light green) form an interaction that might be important for sensing. Essential S580 does not contact the ribosome, whereas Y581 and E588 form H-bonds to R69 and M1 of eS6 (blue), respectively.

E  Mutations in the α-helices of hinge 2 prevent 30S binding while the Y592A/Y593A (C terminus) and L353Y (A-loop in NBSII) exchanges do not influence ribosome binding.

F  70S splitting efficiency normalized to wild type. Hinge 2 mutations Y592A/Y593A, R565E, and S580E display strongly impaired splitting activity. Unspecific ribosome dissociation level as determined in control experiments in the absence of ABCE1 is marked by the dotted line.

G  ATP turnover per ABCE1 is not affected in all tested mutants.

H–J  Overview of ATP coordination in both NBSs and overlay of the two NBSs reveals only slight differences, which cannot elucidate the functional asymmetry. Residues of NBD1 and NBD2 involved in coordination are shown in gold and punch, respectively.

Data Information: In (F) and (G), the mean ± SD of assay triplicates and duplicates are plotted.
Source data are available online for this figure.

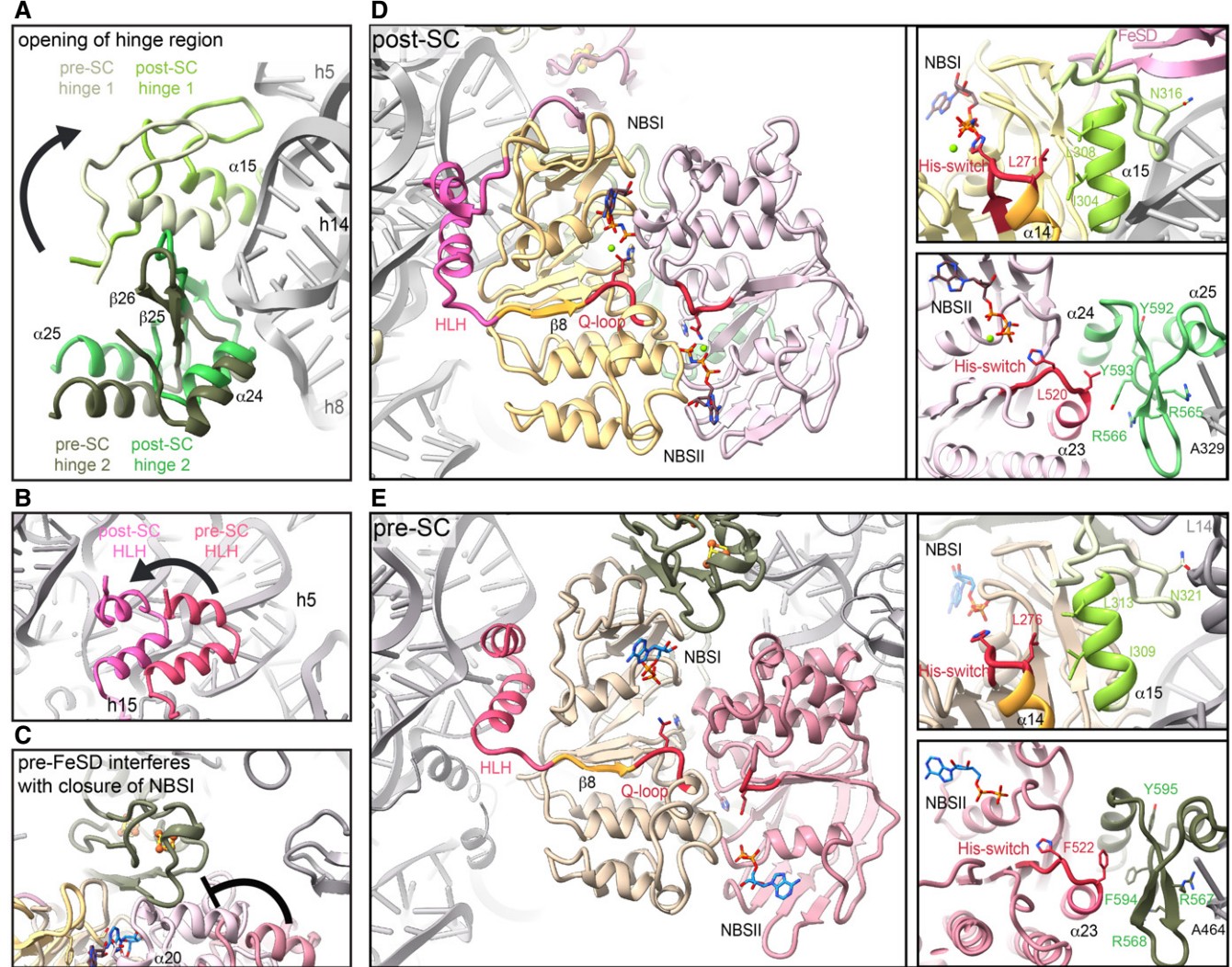

**Figure 4. Hinge regions and HLH sense the ribosome splitting event and allosterically communicate with the NBSs.**

A   Hinge 1 moves away from hinge 2 during transition from pre-SC (cotton) to post-SC (lime), thereby forming new interactions with the ribosome. In contrast, hinge 2 movement from pre- (moss) to post-SC (emerald) does not change the interaction with the ribosome.

B   The HLH motif is displaced from h5 in the pre- (watermelon) to h15 in the post-SC (pink).

C   Positioning of the FeSD (sage) interferes with the closure of NBD2 (blush) in the pre-SC (rose).

D   Possible communication pathways from ribosome binding sites to the NBSs in the post-SC. HLH is connected to the Q-loop of NBSI via β8. I304 of hinge 1 connects to α14 which is adjacent to the His-switch in NBSI. Analogously, hinge 2 binding to the SSU might be communicated via Y593 and R566 to α23 next to the His-switch of NBSII.

E   Interaction pattern of the communication pathways between HLH and hinge 1 to NBSI as well as hinge 2 to NBSII is different in the pre-SC compared to the post-SC.

NBSII (Nürenberg-Goloub *et al*, 2018) and (ii) the lower resolution of this site in cryo-EM studies (Heuer *et al*, 2017). To test this hypothesis, we substituted L353 by a tyrosine, thereby generating a consensus A-loop in NBSII. However, 30S binding, 70S splitting efficiency, and ATPase activity of ABCE1$^{L353Y}$ (Appendix Fig S3) were comparable to wild type (Figs 3E–G and EV4). Consequently, the respective yeast mutation Q363Y had no effect on growth and survival (Figs 2B and EV3C). Thus, the functional asymmetry of ABCE1 may originate from the connection of each NBS to an allosteric regulatory element on the ABCE1 surface, i.e., the FeSD, HLH motif, and hinge regions, rather than from single residues within the ATP-binding pockets.

## Ribosome binding is allosterically communicated to conserved motifs in the NBSs

Ribosome splitting completely alters the interaction pattern of ABCE1 with the ribosome at all contact points excluding the hinge 2 region. Based on the high-resolution structure, we elaborated allosteric communication pathways between the ribosome-ABCE1 interface and the NBSs. In the pre-splitting complex, the FeSD does not interfere with the NBSI semi-open state (Brown *et al*, 2015). However, upon closure, the loop K12-P13-D14 of the FeSD would clash into NBDII, in particular into residues preceding the NBSI signature motif and α20, involving the L453-E454-S455 stretch

(Fig 4C). The movement of NBSI is thus coupled to rearrangements of the FeSD and *vice versa*. Moreover, the flexible HLH motif via β8 is linked to the Q-loop of NBSI (Fig 4D and E). Mutations in the Q-loops strongly affect the ATPase activity of ABCE1 and compromise its function in yeast (Karcher *et al*, 2008; Barthelme *et al*, 2011). As stated above, we observed clear density for Q167 sensing the presence of the γ-phosphate. Additionally, we envision that hinge opening is directly transmitted to the H-loops in both NBSs, which are key motifs in controlling ATPase activity of ABCE1 and other ABC proteins (Zaitseva *et al*, 2005; Barthelme *et al*, 2011; Hurlimann *et al*, 2017). In the post-SC, hinge 1 forms a specific contact to the h5-h15 junction where N316 interacts with G345. Compared to the pre-SC, hinge 1 α15 moves closer toward NBSI and forms a contact with α14, directly adjacent to the H-loop of NBSI (Fig 4D and E). The conserved I304 in α15 points toward α14, allowing a communication between hinge 1 and NBSI. Consistent with this essential function, the corresponding mutation I314E is lethal in yeast (Figs 2B and 4D and E, and EV3C). Similarly, a conserved series of residues communicates ribosome binding from hinge 2 to the H-loop of NBSII. Herein, R565 in hinge 2 senses the h8–h14 junction while R566 and Y593 contact helix α23. Analogously to α14 in NBD1, helix α23 occupies the position adjacent to the H-loop in NBSII (Fig 4D and E). We substituted the conserved Y592 and Y593 by alanine and probed for ABCE1 function. Consistent with the role of Y593 in ribosome sensing without direct contact to rRNA or ribosomal proteins, the 70S splitting ability of ABCE1[Y592A/Y593A] (Appendix Fig S3) is substantially inhibited (Figs 3F and EV4B) while the 30S binding efficiency and ATPase activity are similar to wild type (Figs 3D and E, and EV4A). Additionally, the respective double-mutant Y600A/F601A exhibits a growth defect in yeast (Figs 2B and EV3C). The five-stranded β-sheet harboring the degenerated A-loop in NBSII is in close proximity of hinge 2. Comparing the pre-SC with the post-SC, we observed a conformational change in this region which contributes to ATP occlusion by allowing the hydrophobic stacking of L353 and the adenine base (Fig 3J).

We finally inspected the Walker B/D-loops, which are known to assure transport directionality in the ABC transporter associated with antigen processing (TAP) (Grossmann *et al*, 2014). Notably, the D-loops are, together with the H-loops, already part of the contact interface between the NBDs in the pre-splitting state. This interface drastically alters upon closure of the NBSs, ribosome splitting, and post-SC formation, allowing a multilayered communication network between both sites in addition to the allosteric regulation by the ribosome (Fig 4D and E).

## Discussion

By using an ATPase-deficient mutant of ABCE1 in an *in vitro* ribosome recycling assay, we were able to capture the archaeal post-splitting complex comprising the 30S subunit and ABCE1. Our structure reveals this essential, asymmetric ABC-type protein in a fully nucleotide-occluded state at atomic resolution. Furthermore, the cryo-EM structure allows a prediction of the communication pathways within the post-splitting complex, which we functionally and genetically assessed. Ribosome binding is sensed by the HLH motif and hinge region that opens up during ribosome splitting. This

"hinge opening" modulates the His-switches in both NBSs by altering the contact interface to adjacent α-helices. We observed that NBSI is in an active conformation with all residues needed for catalytic activity in place, i.e., activation of a water molecule for nucleophilic attack on the γ-phosphate (Chen *et al*, 2003; Lammens *et al*, 2011; Hofmann *et al*, 2019). The functional and dynamic asymmetry of the two NBSs (Barthelme *et al*, 2011; Nürenberg-Goloub *et al*, 2018; Gouridis *et al*, 2019) does not arise from incomplete ATP alignment due to a non-canonical A-loop in NBSII, as we confirmed by biochemical and yeast viability studies. In the ABC transporter TAP and its homolog TmrAB, the position of the non-canonical site cannot be switched without compromising the transport function, indicating that additional signals from outside the binding pocket are integrated into the ATPase cycle (Chen *et al*, 2003; Procko *et al*, 2006; Zutz *et al*, 2011). Consistently, we envision an allosteric regulatory network that extends from the ABCE1-ribosome interface into the NBSs. The spatial separation of hinge 1 from hinge 2 is linked to both NBSs and in addition might be a prerequisite for closure of NBSII (Fig 4 and Movie EV1). In agreement, the introduction of mutations disrupting ribosome binding in hinge 1 (R311A in *S.c.*; R301 in *S. solfataricus*) or hinge 2 (R573E, R582E, and S588E in *S.c.*; R565, R574, and S580, in *S. solfataricus*, respectively) compromise ABCE1 function (Karcher *et al*, 2008) (Figs 2B and 3B–D, and EV3C, and EV4). The exchange of G303 in hinge 1 (Appendix Fig S2), located at the contact interface to NBD1, leads to a reduced wing size in *Drosophila melanogaster* (G316D in the *pixie* gene), further highlighting the role of the hinge region for ABCE1 function (Coelho *et al*, 2005). Notably, hinge 1 and hinge 2 occupy a position analogous to the regulatory elements of bacterial ABC importers (Newstead *et al*, 2009; Johnson *et al*, 2012; Chen *et al*, 2013) (Appendix Fig S4), showing that a regulation from this site can be exploited by ABC-type proteins.

Closure of NBSII allosterically activates NBSI, which is consistent with the increased ATPase activity of ABCE1 in the presence of 70S/80S ribosomes and release factors (Pisarev *et al*, 2010; Shoemaker & Green, 2011; Nürenberg-Goloub *et al*, 2018). On a structural level, we assume that NBSII can close prior to NBSI to prime ribosome splitting at the pre-SC (Fig 5). In more detail, the movement of the signature motif toward NBSII is possible when still bound to the 70S/80S ribosomes, since ABCE1 anchors via the hinge 2 region and HLH motif, and none of the mobile parts participate in ribosome binding. Furthermore, 70S/80S are split as soon as both sites occlude Mg$^{2+}$-ATP and switch to the closed conformation (Fig 5), as found within the post-SC (Heuer *et al*, 2017; Nürenberg-Goloub *et al*, 2018; Gouridis *et al*, 2019). During the closing movement, the FeSD is pushed away by NBD2 and, concomitantly, interactions between NBD1, the HLH motif, and the ribosome must be temporarily broken, allowing hinge 1 to move away from hinge 2 (Fig 5). Structurally, separation of the two hinge regions occurs concomitantly with FeSD movement and adoption of the fully closed state of the ABCE1 NBDs. These structural rearrangements may well determine the ribosome splitting rate. Consistently, in the presence of Mg$^{2+}$-AMP-PNP, ABCE1 transiently associates with 30S ribosomes within 5 s, while closure of NBSII takes app. 7 min and stabilizes the post-SC (Gouridis *et al*, 2019).

Remarkably, translation termination is a slow event. Several ribosome profiling studies showed a high enrichment of reads

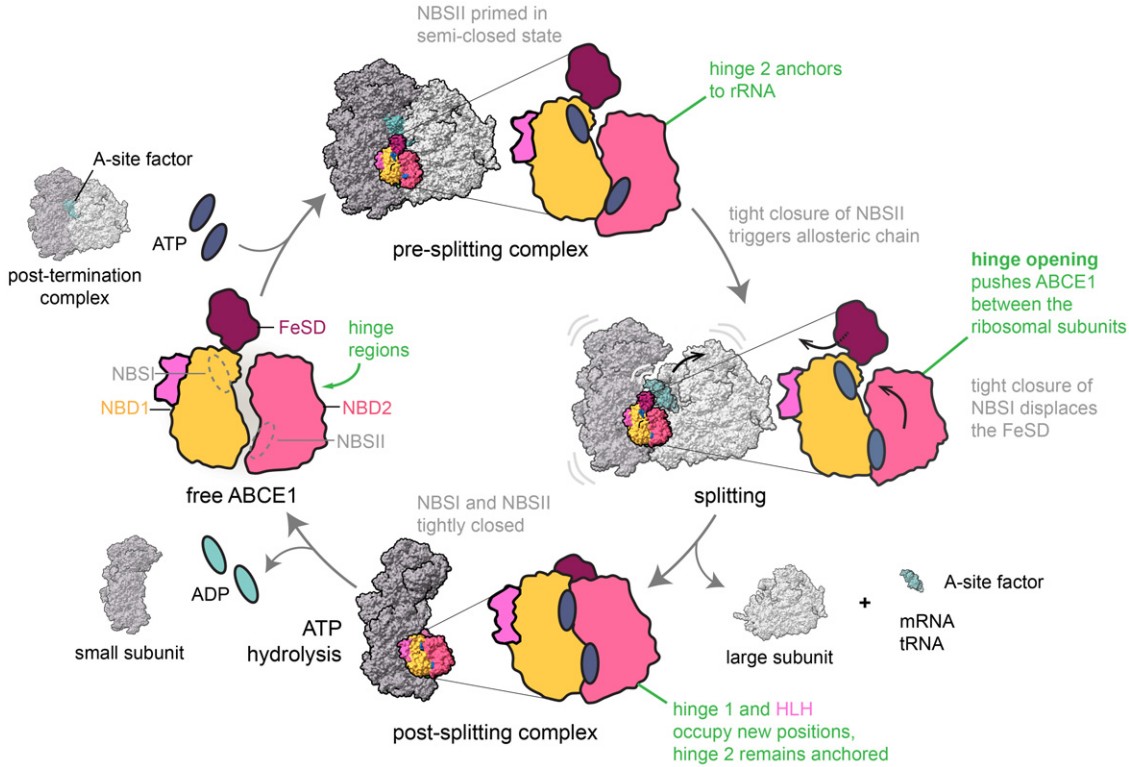

**Figure 5. Model for ribosome splitting by ABCE1.**

ABCE1 binds to 70S/80S ribosomes containing mRNA, tRNA in the P-site (not shown), and an A-site factor (a/eRF1 after canonical termination; a/e Pelota during stalled ribosome recognition) to form pre-splitting complexes. Here, NBSII is primed in a semi-closed state and anchored to ribosomal RNA via hinge 2. ATP occlusion and tight closure of NBSII triggers an allosteric chain within ABCE1 leading to a tight closure of NBSI. Consequently, the FeSD is displaced and the parallel hinge opening rearranges ABCE1 in the ribosomal subunit cleft. Thereby, the subunits are split apart and the FeSD is repositioned at h44. During and/or after the splitting process, the A-site factor dissociates and mRNA and tRNA are recycled (not shown). At the post-SC, ABCE1 occludes two ATP molecules in the NBSs. ATP hydrolysis is a prerequisite for NBS opening and dissociation of ABCE1 from the SSU. Black arrows indicate domain movements within ABCE1.

indicating a high occupancy of ribosomes on stop codons (Andreev *et al*, 2017). Moreover, a significant population of ABCE1-containing termination complexes was found in native polysomes, along with translating ribosomes (Behrmann *et al*, 2015). Similarly, the half-life of ribosomes stalled during translation and rescued by the Pelota/Hbs1/ABCE1 system is supposedly long. In light of this, it makes sense that ribosome splitting is regulated and coordinated by the action of the intrinsically slow NBSII. Slow closure of NBSII could ensure correct engagement within the pre-splitting complex, and slow ATP hydrolysis could determine the dwell time of ABCE1 after splitting to prevent premature re-association with large ribosomal subunits, or coordinate downstream events such as translation initiation and/or tRNA/mRNA recycling. In this context, the question remains open as to how ATPase activity and thus the 30S/40S dissociation is modulated (Fig 5). Here, external factors, e.g., components of the initiation machinery, might play a direct or indirect role in communicating conformational rearrangements during pre-initiation complex formation into the NBSs of ABCE1 to trigger its release. In particular, and possibly by modulating its ATPase activity, the non-essential eukaryotic eIF3j subunit (Hcr1 in *S.c.*) assists ABCE1 in ribosome recycling, and thereby may also promote post-SC disassembly (Young & Guydosh, 2019). In the

future, the precise role of ABCE1 in initiation will need to be elucidated to complete the translation cycle for Eukarya and Archaea.

# Material and Methods

### Protein purification

Construction of the pSA4 plasmids for recombinant expression of ABCE1, aRF1, aPelota, and aIF6 from *S. solfataricus* in *E. coli* was described previously (Barthelme *et al*, 2007, 2011). All proteins were expressed, purified, and stored as previously described (Nürenberg-Goloub *et al*, 2018). Protein quality was assured by SDS–PAGE and size exclusion chromatography (Superdex 200 Increase 3.2/300, GE Healthcare) in SEC buffer (20 mM Tris–HCl pH 7.5, 150 mM NaCl, 2 mM β-mercaptoethanol) at 4°C recording absorption at 280 and 410 nm to monitor FeSD cluster integrity.

### Ribosome purification

Frozen cell pellets from *T. celer* were purchased from the Centre of Microbiology and Archaea, University of Regensburg, Germany.

Cell pellets were resuspended in 2.5× volume S30 buffer (10 mM Tris–HCl pH 7.5, 60 mM KOAc, 14 mM MgCl$_2$, 1 mM dithiothreitol (DTT)) and lysed using a Branson Sonifier. Cell debris was removed by centrifugation 2 × 30 min at 34,000 $g$ and 4°C. The supernatant was loaded on a high-salt sucrose cushion (10 mM Hepes-KOH pH 7.5, 1.1 M sucrose, 1 M NH$_4$Cl, 10.5 mM Mg (OAc)$_2$, 0.1 mM EDTA, 4 mM β-mercaptoethanol), and ribosomes were pelleted at 200,000 $g$ for 15 h at 4°C. For 70S preparation, pelleted ribosomes were resuspended in S30 buffer and gradient purified (10–40% (w/v) sucrose, S30 buffer) for 14 h at 68,000 $g$. Fractions were collected using a Piston Gradient Fractionator (Biocomp) recording the A$_{254}$ profile. The buffer of 70S containing fractions was exchanged to TrB25 (56 mM Tris–HCl pH 8.0, 250 mM KOAc, 80 mM NH$_4$OAc, 25 mM MgCl$_2$, 1 mM DTT) via Econo-Pac 10DG Desalting Columns (Bio-Rad), and 70S were concentrated using a 100K Amicon Ultra (Merck). For 30S purification (for 30S binding assays), high-salt sucrose cushion pelleted ribosomes were resuspended in buffer A30 (10 mM Hepes-KOH pH 7.5, 100 mM NH$_4$Cl, 10.5 mM Mg(OAc)$_2$, 0.1 mM EDTA, 4 mM β-mercaptoethanol) and loaded onto a HiPrep 16/60 Sephacryl S-400 HR size exclusion chromatography column (GE Healthcare). Ribosome fractions were collected and again pelleted through a low magnesium sucrose cushion in buffer A30 (2.5 mM Mg(OAc)$_2$) for subunit dissociation. Ribosomes were resuspended in S30 buffer (with 2.5 mM Mg(OAc)$_2$ instead of MgCl$_2$) and gradient purified. 30S fractions were pooled, the buffer exchanged to S30 and concentrated as before.

## Assembly of the post-splitting complex for cryo-EM

To mimic the physiological translation cycle, post-splitting complexes were generated by splitting of 1 nmol purified 70S ribosomes from *T. celer* by ABCE1$^{IIEA}$ (8 µM), aPelota, and aRF1 (5 µM each) in the presence of 0.5 mM AMP-PNP in 50 mM HEPES-KOH pH 7.5, 30 mM KCl, 10 mM MgCl$_2$, and 2 mM DTT at 65 °C for 15 min. Samples were chilled on ice and cross-linked with 1% (v/v) formaldehyde for 30 min on ice. Higher molecular weight aggregates were removed for 15 min at 16,100 $g$ and 4 °C. Samples were loaded onto 10–30% (w/v) sucrose density gradient in 50 mM HEPES-KOH pH 7.5, 30 mM KCl, 0.5 mM MgCl$_2$, and 2 mM DTT, and ribosomal particles were separated by centrifugation for 13.5 h at 78,000 $g$ and 4°C in a SW40 rotor (Beckman Coulter Life Sciences). Gradients were fractionated into 0.3 ml using Piston Gradient Fractionator (Biocomp Instruments) while recording A$_{254}$. Fractions containing 30S were pooled, and the sucrose was removed by Sephadex G-25 gravity flow size exclusion columns (GE Healthcare). Ribosomes were diluted to concentrations of 50–70 nM (based on OD$_{260}$) for quality control by negative stain EM. Samples were vitrified immediately.

## 70S splitting assay

7.5 pmol *T. celer* 70S were split using ABCE1, aRF1, aPelota, and aIF6 (75 pmol each) in the presence of 22.5 nmol AMP-PNP in S30 buffer at 65°C for 15 min. Higher molecular weight aggregates were removed for 10 min at 16,100 $g$ and 4°C. Samples were analyzed via 10–40% (w/v) sucrose density gradient in S30 buffer as described. Splitting efficiency was calculated as the ratio of 50S peak

area to 70S peak area of the A$_{254}$ gradient profile using OriginPro 2018 (OriginLab) and normalized to the mean value of wild-type ABCE1. Splitting experiments were performed at least three times per ABCE1 variant; bars show mean ± SD value.

## 30S binding assay

17.5 pmol *T. celer* 30S were incubated with 8.5 pmol ABCE1 in the presence of AMP-PNP, or ADP (8.5 nmol each), or in the absence of any nucleotide in S30 buffer for 10 min at 65°C. Higher molecular weight aggregates were removed for 10 min at 16,100 $g$ and 4°C. Samples were loaded onto a 10–40% (w/v) sucrose density gradient in S30 buffer, as described. 0.5-ml fractions were collected, precipitated overnight at –20°C in 2× volume acetone, and pelleted for 1 h at 16,100 $g$ and 4 °C. The pellet was resuspended in SDS loading dye and analyzed by SDS–PAGE and immunoblotting. All ABCE1 variants contained a C-terminal His$_6$ tag and were detected using rabbit anti-His (ab1187, Abcam) and goat anti-rabbit (AP307P, Merck) antibodies. Binding assays were performed once per ABCE1$^{variant}$. The gradient profiles shown are representative for the respective nucleotide condition.

## ATPase assay

ATPase activity was measured using a Malachite Green-based assay (adapted from (Baykov *et al*, 1988). Samples were measured at least in duplicates. 1–2 µM ABCE1 was incubated with 2 mM ATP in ATPase buffer (10 mM Hepes pH 7.5, 150 mM NaCl, 2.5 mM MgCl$_2$) for 8 min at 80°C in a total volume of 25 µl. The reaction was stopped by addition of 175 µl ice-cold 20 mM H$_2$SO$_4$. 50 µl Malachite Green working solution (2 ml conc. Malachite Green solution (60 ml H$_2$SO$_4$ in 300 ml H$_2$O with 0.44 g Malachite Green), 40 µl Tween-20 (10% v/v) and 550 µl Na$_2$MoO$_4$) was added per sample and incubated for 2–5 min at room temperature. A$_{620}$ was recorded in a CLARIOstar plate reader (BMG Labtech). Bar diagrams represent mean ± SD of two (ABCE1$^{S580E}$), four (ABCE1$^{L353Y}$) or three (all other ABCE1 variants) independent experiments.

## Yeast plasmid shuffling assay

*In vivo* function of ABCE1 mutants was checked as previously described (Heuer *et al*, 2017). The haploid yeast strain CEN.MG1-9B (*MATa his3Δ1 leu2-3,112 trp1-289 MAL2-8$^C$ SUC2 ura3-52 rli1::KanMX4* + pRS426-ABCE1) was generated in which the essential *ABCE1* gene (*RLI1*) was deleted by *KanMX4* and substituted by pRS426-ABCE1 [*URA3*] expressing wild-type *ABCE1* under the control of the endogenous promoter. CEN.MG1-9B strain was transformed with pRS423-ABCE1 [*HIS3*] plasmid coding for wt and mutated *ABCE1* and with empty vector pRS423 as negative control and selected on -HIS. If such a strain harboring both plasmids was grown on medium containing 5-FOA, the pRS426-ABCE1 [*URA3*] plasmid is lost by counter-selection as the *URA3* gene product converts 5-FOA to a toxic compound. Consequently, the strain was prone to survive only in the presence of pRS423-ABCE1. Growth and survival were checked by growth studies in a serial dilution assay over 2–3 days. Data in Figs 2B and EV4C are representative for a set of two independent experiments.

## Cryo-EM analysis

For the archaeal post-SC, the sample was applied to 2-nm pre-coated Quantifoil R3/3 holey carbon-supported grids and vitrified using a Vitrobot mark IV (FEI). Data were collected on a TITAN KRIOS™ cryo-TEM (Thermo Fisher) equipped with a Falcon III chip enhanced Falcon II direct detector at 300 keV under low-dose conditions of approximately 25 e⁻/Å² for 10 frames in total, and a defocus range of −1.1 to −2.3 μm. Magnification settings resulted in a pixel size of 1.084 Å per pixel. Original image stacks were summed and corrected for drift and beam-induced motion at the micrograph level by using MotionCor2 (Zheng *et al*, 2017). The contrast transfer function (CTF) estimation of each micrograph was performed with Gctf (Zhang, 2016).

## Data processing

The ABCE1-30S data set was processed, unless otherwise stated, following the standard workflow using RELION 2 and 3 (Kimanius *et al*, 2016; Zivanov *et al*, 2018). After particle picking with GAUTO-MATCH (http://www.mrc-lmb.cam.ac.uk/kzhang/) and 2D classification, particles were subjected to a thorough 3D classification regimen. About 97% of all particles contained ABCE1 stably bound to the small ribosomal subunit. Different conformational states of the ribosome 30S head were separated and a homogeneous class with 293.010 particles was selected for further refinement. First, the particles of this class were CTF-corrected and refined to an overall resolution of 2.8 Å after post-processing. A focused refinement on the head and ABCE1 could improve the local resolution of the structure.

## Model building

The molecular model of the small ribosomal subunit was built using the 70S model of *P.fu.* [4V6U (Armache *et al*, 2013), 5JBH (Coureux *et al*, 2016)]. After rigid-body fitting of the 30S into the density, the sequence was manually changed to *T. celer* and modeled into the cryo-EM density using Coot (version 0.8.9.1) (Emsley & Cowtan, 2004). The sequences were taken from the *T. celer* Vu 13 = JCM 8558A genome, available at NCBI. A previously unidentified protein could be modeled by building the sequence *de novo* into the density. The *T. celer* genome was searched for characteristic sequence motifs of the protein taking the approximate size of the protein into consideration. An initial model of ABCE1 was generated using Phyre2 (Kelley *et al*, 2015). The model of *S.c.* ABCE1 (Heuer *et al*, 2017) was used as a template, and the resulting model was manually refined in Coot. After Phenix refinement (Adams *et al*, 2010), the models and maps of 30S head, body, and ABCE1 were combined and refined again. Cryo-EM structures and models were displayed using UCSF Chimera (Pettersen *et al*, 2004) and ChimeraX (version 0.8; Goddard *et al*, 2018).

## Data availability

The cryo-EM density maps of the archaeal 30S ribosome and ABCE1 have been deposited in the Electron Microscopy Data Bank under accession number EMD-10519 (https://www.ebi.ac.uk/pdbe/entry/emdb/EMD-10519) (see Table 1). Atomic coordinates for the

**Table 1. Data collection, refinement, and validation statistics.**

| | 30S-ABCE1 (EMD-10519, PDB ID 6TMF) |
|---|---|
| Data collection | |
| Voltage (kV) | 300 |
| Electron exposure (e–/Å²) | 25 |
| Defocus range (μm) | −1.1 to −2.3 |
| Pixel size (Å) | 1.084 |
| Symmetry imposed | C1 |
| Refinement | |
| Particle images (no.) | 293 010 |
| Map resolution (Å) (overall/30S head/ABCE1) | 2.8/2.8/3.0 |
| FSC threshold | 0.143 |
| Map sharpening B factor (Å²) (overall/30S head/ABCE1) | −117.8/−128.8/−151.9 |
| Model composition | |
| Correlation coefficient (%; Phenix) | 0.85 |
| Initial model used (PDB codes) | 5JBH, ABCE1: 5LL6 (chain h) |
| Non-hydrogen atoms | 65 449 |
| Protein residues | 4 172 |
| RNA bases | 1 485 |
| R.m.s. deviations | |
| Bond lengths (Å) | 0.017 (30) |
| Bond angles (°) | 1.208 (44) |
| Validation | |
| MolProbity score | 1.79 |
| Clash score | 4.66 |
| Rotamer outliers (%) | 0.73 |
| Ramachandran plot | |
| Favored (%) | 89.98 |
| Allowed (%) | 9.51 |
| Disallowed (%) | 0.51 |
| Validation RNA | |
| Correct sugar pucker (%) | 98 |
| Good backbone conf. (%) | 80 |

atomic models have been deposited in the Protein Data Bank under accession number PDB ID 6TMF (https://doi.org/10.2210/pdb 6TMF/pdb). Correspondence and requests for materials should be addressed to R.T. (tampe@em.uni-frankfurt.de) or R.B. (beck-mann@genzentrum.lmu.de).

**Expanded View** for this article is available online.

## Acknowledgements

The authors thank Simon Trowitzsch, Lukas Susac, Jingdong Cheng, and Michael Ameismeier for discussions; Charlotte Ungewickell and Susanne Rieder for technical assistance; Lukas Kater for support with processing and the pre-processing pipeline of the cryo-EM data; and Petr Tesina for help with final model refinements in Phenix. E.N.G. was supported by the Christiane Nüsslein-

Volhard Foundation, L'Oréal, and the United Nations Educational, Scientific and Cultural Organization (UNESCO). H.K. is supported by a DFG fellowship through the Graduate School of Quantitative Bioscience Munich (QBM). The German Research Foundation (DFG) SFB 902 "Molecular mechanisms of RNA-based regulation" (to R.T.), TRR174 "Spatiotemporal dynamics of bacterial cells" (to R.B.) and FOR 1805 (to R.B.) funded this work.

## Author contributions

EN-G, HK, HH, TB, RB, and RT designed the study. EN-G and HH developed the preparation of the post-splitting complex. EN-G, HK, HH, and AH optimized the sample preparation for cryo-EM. EN-G, HK, HH, and AH prepared the EM samples. HK and OB collected and HK processed the cryo-EM data. HK built and refined the model. HK, TB, EN-G, HH, RT and RB analyzed and interpreted the structures. HH and EN-G performed all functional assays. EN-G and PK conducted the genetic analysis in yeast. EN-G, HK, TB, HH, RB, and RT wrote the manuscript with contributions from all authors. RT initiated the project.

## Conflict of Interest

The authors declare that they have no conflict of interest.

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
