## [Review Process File · The EMBO Journal]

Molecular analysis of the ribosome recycling factor ABCE1 bound to the 30S post-splitting complex

Elina Nürenberg-Goloub, Hanna Kratzat, Holger Heinemann, André Heuer, Peter Kötter, Otto Berninghausen, Thomas Becker, Robert Tampe, and Roland Beckmann

Review timeline:	Submission date:	20th Oct 19
	Editorial Decision:	18th Nov 19
	Revision received:	12th Dec 19
	Editorial Decision:	13th Jan 20
	Revision received:	17th Jan 20
	Accepted:	21st Jan 20

Editor: Stefanie Boehm

Transaction Report:

1st Editorial Decision

18th Nov 19

Thank you for submitting your manuscript on the 30S-ABCE1 post-splitting complex for consideration by The EMBO Journal. We have now received two referee reports on your study, which are included below for your information.

As you will see, both referees are overall positive and acknowledge the interest and potential contribution to the field. Nonetheless they still raise some issues that would need to be addressed or discussed in a revised version of the manuscript. Should you be able to adequately do so, we would be happy to consider this study further for publication. Therefore I would like to invite you to prepare and submit a revised manuscript. Please note that it is our policy to allow only a single round of major revision and that it is therefore important to clarify all key concerns raised at this stage.

REFERE REPORTS

Referee #1:

The Beckmann and Tampé groups have previously collaborated to use cryoEM to determine the structure and location of the recycling factor ABCE1 on the yeast 40S ribosomal subunit, which together recapitulate elements of the complex that is formed after ABCE1-mediated splitting of ribosomes (Heuer et al., 2017). In the present report by Nürenberg-Goloub et al., these two groups used ribosomes from *Thermococcus celer* and factors from *Saccharolobus solfataricus* to determine the cryoEM structure of the analogous archaeal 'post-splitting' complex. The significantly higher resolution of the present complex (average 2.8Å vs the previous 3.9Å) provided support for the proposed conservation of this mechanism in eukaryotes and archaea, and revealed new details of the interaction between domains of ABCE1 and between ABCE1 and the small (30S) ribosomal subunit. These new data enabled the authors to refine their model for the mechanism of ABCE1-

mediated ribosomal splitting, to test aspects of it using yeast genetics (to characterize growth and viability of mutant strains) and biochemistry (ribosomal binding, ribosomal splitting and ATPase activity of purified mutant proteins), and to interpret existing mutational analyses (e.g. Kärcher et al., 2006). Comparison with the conformation of ABCE1 in the post-termination/pre-splitting complex allowed the authors to propose how ribosomal binding is allosterically communicated to the nucleotide-binding sites. This work is expertly done and in general clearly presented, and therefore constitutes a valuable addition to the field.

SPECIFIC COMMENTS

1. It is impossible to understand why the authors included aRF1 in splitting reactions that contained aPelota and vacant 70S ribosomes (p4, line 19; p13, line 28; p14, line 5). This must be clarified. It is also absolutely unclear how and why the ATPase assay was done in the absence of Mg²⁺ (see the composition of the ATPase buffer).

2. The authors' model for ABCE1-mediated ribosomal splitting (Figure 5) is a significant oversimplification and will confuse non-experts. The authors should clearly explain that it omits many key features of the process, such as (a) the presence of eRF1/Pelota, mRNA and deacylated P-site tRNA in native pre-splitting complexes, (b) the continued presence of mRNA and deacylated P-site tRNA in the post-splitting complex. The omission of eRF1/Pelota is particularly troubling because they are essential for splitting.

3. The process of stop codon-independent splitting of ribosomes (p3, lines 16-17) is poorly explained and will not be comprehensible to non-experts.

4. p13, lines 2-3. The description of protein purification is inadequate, because the cited reference does not describe the aPelota expression vector, and does not describe the sequences that were cloned into pSA4 e.g. by reference to GenBank accession numbers. Did the storage-G150 buffer contain glycerol and beta-mercaptoethanol, as in the 2018 Nürenberg-Goloub et al. paper?

5. Typographical errors.

p6, line 9. "30S" refers to a ribosomal subunit rather than to the entire ribosome.

p14, line 16. The authors should change text from French to English.

p15, line 19. I assume that the authors mean 'regimen' rather than 'regiment'.

Referee #2:

In this interesting manuscript, the cryo-EM structure of an archaeal 30S ribosome in complex with the ribosome recycling factor ABCE1 is presented. The *in vitro* reconstitution procedure used led to the enrichment of a post-splitting complex, in which ABCE1 is bound to the 30S subunit and adopts a conformation that is incompatible with the existence of a 70S particle. Together with earlier structures of the pre-splitting complex and of ABCE1 in isolation, this work provides a key missing structural snapshot to help us understand the mechanism by which ABCE1 splits the post-termination ribosome. Specifically, it reveals how the ribosome induces conformational changes within ABCE1 that lead to the closure of its nucleotide-binding sites and turn on its ATPase activity.

The results presented in this work make a key contribution to an important and only partially understood aspect of the translational cycle of archaea and eukaryotes. The structural and biochemical experiments are performed to high standards, the data are clearly presented and the conclusions are fully justified based on the available data.

As a result, I only have the following minor comments:

- On p.6 (second paragraph) and Fig. 2A, there are a number of interactions that need to be reexamined, namely:

- Residue N17 cannot form a salt bridge as stated in the text. Perhaps the authors were referring to a possible hydrogen bond with the 2'OH of G1368?

- E19 is said to take part in a salt bridge, but this could only occur with a positively charged partner.

However, this does not appear to be the case in Fig. 2A. Could the authors specify the exact nature of the interaction that is observed?

- R144 and K146 appear to make salt bridges with the backbone phosphates, but these interactions are not mentioned in the text.

- E76 of uS12 is mentioned in the text, but this appears to be a typo since it is clearly Q76 in Fig. 2A.

- In Fig. 3, A138 is not labeled in panels A and D despite being mentioned on p. 7, third paragraph.
- Similarly, N316 and S580 are not labeled in Fig. 3A, despite being mentioned at the bottom of page 7.
- In Fig. 3H, it would be useful to include density for the ligand since the text mentions that the high-resolution allows the identification of the bound nucleotide analogue and an Mg²⁺ ion.
- In Fig. 4J, G216 should be labeled
- In Fig. 5, Shouldn't the arrows reflecting domain movements in ABCE1 during splitting go in the opposite direction? At least this is what I would think based on the accompanying movie.

1st Revision - authors' response

12th Dec 19

Point-to-point reply to the two referees:

Referee #1:

*The Beckmann and Tampé groups have previously collaborated to use cryoEM to determine the structure and location of the recycling factor ABCE1 on the yeast 40S ribosomal subunit, which together recapitulate elements of the complex that is formed after ABCE1-mediated splitting of ribosomes (Heuer et al., 2017). In the present report by Nürenberg-Goloub et al., these two groups used ribosomes from *Thermococcus celer* and factors from *Saccharolobus solfataricus* to determine the cryoEM structure of the analogous archaeal 'post-splitting' complex. The significantly higher resolution of the present complex (average 2.8 Å vs the previous 3.9 Å) provided support for the proposed conservation of this mechanism in eukaryotes and archaea, and revealed new details of the interaction between domains of ABCE1 and between ABCE1 and the small (30S) ribosomal subunit. These new data enabled the authors to refine their model for the mechanism of ABCE1-mediated ribosomal splitting, to test aspects of it using yeast genetics (to characterize growth and viability of mutant strains) and biochemistry (ribosomal binding, ribosomal splitting and ATPase activity of purified mutant proteins), and to interpret existing mutational analyses (e.g. Kärcher et al., 2006). Comparison with the conformation of ABCE1 in the post-termination/pre-splitting complex allowed the authors to propose how ribosomal binding is allosterically*

communicated to the nucleotide-binding sites. This work is expertly done and in general clearly presented, and therefore constitutes a valuable addition to the field.

Reply: We kindly thank the reviewer for the very positive evaluation of our work and her/his helpful critical comments on our study.

SPECIFIC COMMENTS

1. It is impossible to understand why the authors included aRF1 in splitting reactions that contained aPelota and vacant 70S ribosomes (p4, line 19; p13, line 28; p14, line 5). This must be clarified. It is also absolutely unclear how and why the ATPase assay was done in the absence of Mg²⁺ (see the composition of the ATPase buffer).

Reply: We apologize for this unclarity. It is likely that the purified *T. celer* ribosomes still contain a population that harbors mRNA/tRNAs and nascent peptides since these ribosomes were not puromycin treated. Moreover, it is known from diverse ribosome profiling experiments that a large population of ribosomes is stalled on stop codons. In order to assure that all species of 70S ribosomes were captured, we added aRF1 on top of aPelota in splitting reactions to obtain the highest splitting efficiency.

We added a short explanation to the main text of the revised manuscript. It now reads as follows: “To obtain archaeal post-SCs, we actively split isolated native *Thermococcus celer* (*T.c.*) 70S ribosomes using recombinant ABCE1, aRF1, and aPelota from the related archaeon *Saccharolobus solfataricus* (*S.s.*), thus ensuring to resemble the cellular recycling route for all ribosomes present in the native mixture: ribosomes with the A-site occupied by a stop codon (aRF1), a sense codon (e.g. in stalled ribosomes) or vacant ribosomes (aPelota).“

Concerning the absence of magnesium in our ATPase assay: Mg²⁺ was of course present in the ATPase buffer (2.5 mM). We corrected the composition of the ATPase buffer in the methods part of the revised manuscript.

2. The authors' model for ABCE1-mediated ribosomal splitting (Figure 5) is a significant over-simplification and will confuse non-experts. The authors should clearly explain that it omits many key features of the process, such as (a) the presence of eRF1/Pelota, mRNA and deacylated P-site tRNA in native pre-splitting complexes, (b) the continued presence of mRNA and deacylated P-site tRNA in the post-splitting complex. The omission of eRF1/Pelota is particular troubling because they are essential for splitting.

Reply: We thank the reviewer for this criticism. We are aware of the oversimplification of the recycling/splitting process which has been detailed in other recent publications (e.g. Nürenberg-Goloub & Tampé, 2019, *Biological Chemistry*). In this manuscript, we intended to focus only on our key findings of the current work. Yet, we agree with the referee that leaving out key players of the recycling process is confusing to the reader and therefore added the A-site factor (a/eRF1 or a/ePelota) to all our structural thumbnails. However, we still prefer to omit mRNA and tRNAs because their recycling is not within the scope of this paper and would overcomplicate the matter. This is because the association of mRNA and tRNAs with 40S and/or 60S differs in stop codon-dependent and non-canonical ribosome recycling pathways (e.g. peptidyl tRNA remains bound to the 60S after Pelota-mediated splitting). In addition, recycling of mRNA/tRNA after canonical splitting is mediated by the eIF2D/MCTS1/DENR system. Thus, to maintain the focus on ABCE1, we mention mRNA and tRNA as key features of the recycling process only in the revised figure legend. Furthermore, we extended the introduction, which now describes the composition of the different complexes more in detail (see also point 3.).

The revised legend for Figure 5 now reads as follows: “ABCE1 binds to 70S/80S ribosomes containing mRNA, tRNA in the P-site (not shown), and an A-site factor (a/eRF1 after canonical termination; a/e Pelota during stalled ribosome recognition) to form pre-splitting complexes. Here, NBSII is primed in a semi-closed state and anchored to ribosomal RNA *via* hinge 2. ATP occlusion and tight closure of NBSII triggers an allosteric chain within ABCE1 leading to a tight closure of NBSI. Consequently, the FeSD is displaced and the parallel hinge-opening rearranges ABCE1 in the ribosomal subunit cleft. Thereby, the subunits are split apart and the FeSD is repositioned at h44. During and/or after the splitting process, the A-site factor dissociates and mRNA and tRNA are recycled (not shown). At the post-SC, ABCE1 occludes two ATP molecules in the NBSs. ATP hydrolysis is a prerequisite for NBS opening and dissociation of ABCE1 from the SSU. Black arrows indicate domain movements within ABCE1.”

3. The process of stop codon-independent splitting of ribosomes (p3, lines 16-17) is poorly explained and will not be comprehensible to non-experts.

Reply: As already mentioned in point 2, we extended the introduction of the revised manuscript and included additional information on ribosome recycling after stop

codon-independent non-canonical splitting as well as on the composition of pre- and post-SCs. We also provided additional literature.

In the introduction, it now reads as follows: “Herein, the conserved and essential ATP-binding cassette (ABC)-type twin-ATPase ABCE1 plays the key role for Archaea and Eukarya (Barthelme et al., 2011, Pisarev et al., 2010, Shoemaker & Green, 2011). It recycles canonical 70S/80S post-termination complexes (post-TCs) after stop codon-dependent termination and non-canonical post-TCs during mRNA surveillance and resumption of translation after cellular stress. In both cases, a decoding A-site factor (archaeal/eukaryotic release factor 1 (a/eRF1) or its homologue a/ePelota, respectively) is delivered to the ribosomal A-site by a translational GTPase (aEF1/eRF3 or aEF1/Hbs1, respectively) and forms an interaction platform for ABCE1 to establish the 70S/80S pre-splitting complex (pre-SC) (Becker et al., 2012, Brown et al., 2015, Preis et al., 2014, Shao et al., 2016). In concert with the A-site factor, ABCE1 splits the pre-SC into the small (SSU) and large (LSU) ribosomal subunit. In Eukarya, other components of the post-TC stay associated with the ribosomal subunits and are subsequently recycled by additional factors (Pisarev et al., 2010, Skabkin et al., 2010). Canonical termination, which includes peptide release by eRF1, yields 40S-mRNA-deacylated tRNA complexes and free 60S subunits whereas ribosome recycling of non-canonical post-TCs in the presence of Pelota results in 40S-mRNA and 60S-peptidyl-tRNA complexes due to Pelota’s incapacity to release peptides. Moreover, Pelota/Hbs1/ABCE1 not only acts in the splitting of stalled (Shoemaker & Green, 2011), but also vacant (van den Elzen et al., 2014), and newly synthesized ribosomes (Strunk et al., 2012). Immediately after splitting, an ABCE1-bound 30S/40S post-splitting complex is formed (Heuer et al., 2017, Kiosze-Becker et al., 2016), in which ABCE1 may remain for a defined time span (Gouridis et al., 2019, Nürenberg-Goloub et al., 2018) to prevent re-association of the LSU (Heuer et al., 2017).”

4. p13, lines 2-3. The description of protein purification is inadequate, because the cited reference does not describe the aPelota expression vector, and does not describe the sequences that were cloned into pSA4 e.g. by reference to GenBank accession numbers. Did the storage-G150 buffer contain glycerol and beta-mercaptoethanol, as in the 2018 Nürenberg-Goloub et al. paper?

Reply: We apologize for this confusion. In the revised methods part, we included additional information on the origin of the mentioned plasmids. Notably, these plasmids were constructed and previously published by the Tampé group. Expression, purification, and storage of the proteins was in fact done exactly as in

Nürnberg-Goloub *et al.* 2018, including the composition of all buffers. The SEC was performed in a buffer without glycerol, not in Storage-G₁₅₀ buffer. We corrected the information in the revised Materials & Methods part accordingly.

5. Typographical errors.

p6, line 9. "30S" refers to a ribosomal subunit rather than to the entire ribosome.

p14, line 16. The authors should change text from French to English.

p15, line 19. I assume that the authors mean 'regimen' rather than 'regiment'.

Reply:

We corrected '30S ribosome' to '30S ribosomal subunit'.

We rearranged the sentence to remove the French preposition 'à'.

Indeed, we meant 'regimen' and apologize for the typo.

Referee #2:

In this interesting manuscript, the cryo-EM structure of an archaeal 30S ribosome in complex with the ribosome recycling factor ABCE1 is presented. The in vitro reconstitution procedure used led to the enrichment of a post-splitting complex, in which ABCE1 is bound to the 30S subunit and adopts a conformation that is incompatible with the existence of a 70S particle. Together with earlier structures of the pre-splitting complex and of ABCE1 in isolation, this work provides a key missing structural snapshot to help us understand the mechanism by which ABCE1 splits the post-termination ribosome. Specifically, it reveals how the ribosome induces conformational changes within ABCE1 that lead to the closure of its nucleotide-binding sites and turn on its ATPase activity.

The results presented in this work make a key contribution to an important and only partially understood aspect of the translational cycle of archaea and eukaryotes. The structural and biochemical experiments are performed to high standards, the data are clearly presented and the conclusions are fully justified based on the available data.

As a result, I only have the following minor comments:

Reply: We kindly thank the reviewer for the very positive evaluation of our work and the helpful comments on our study.

- On p.6 (second paragraph) and Fig. 2A, there are a number of interactions that need to be reexamined, namely:

Reply: We thank the reviewer for his detailed inspection of our structural figures. We carefully went through all interactions again and thoroughly checked if they are correctly displayed and described.

- Residue N17 cannot form a salt bridge as stated in the text. Perhaps the authors were referring to a possible hydrogen bond with the 2'OH of G1368?

Reply: We realized that we were imprecise in our first manuscript. The referee is right that residue N17 of ABCE1 forms a hydrogen bond instead of a salt bridge. The hydrogen bond is formed with the 2'OH of U1427, which we now labeled in revised Figure 2A.

- E19 is said to take part in a salt bridge, but this could only occur with a positively charged partner. However, this does not appear to be the case in Fig. 2A. Could the authors specify the exact nature of the interaction that is observed?

Reply: Again, the referee is correct. Residue E19 of ABCE1 forms a hydrogen bond to 2'OH of G1426 and phosphate of U1427. We adjusted the text accordingly.

- R144 and K146 appear to make salt bridges with the backbone phosphates, but these interactions are not mentioned in the text.

Reply: As the perspective in Fig. 2A suggests, residues R144 and K146 are indeed in close vicinity to the mentioned residues of rRNA h15. We measured the distances of all possible salt-bridge interactions. The distances between K146 and the phosphate group of U353 is 4.4 Å; the distance between K146 and the phosphate group of C352 is 5.2 Å; and the distance between R144 and the phosphate group of G380 is 5.9 Å. Thus, all distances are too long as to account for a salt bridge (< 4 Å), and we therefore do not mention these interactions in the revised text. To that end, we also deleted the mentioning of the salt bridge between R144 and G380 later in the manuscript.

- E76 of uS12 is mentioned in the text, but this appears to be a typo since it is clearly Q76 in Fig. 2A.

Reply: We corrected Q76 in the text.

In summary, the revised paragraph describing Figure 2A now reads as follows: “The majority of interactions are formed by salt bridges and hydrogen bonds established between conserved residues in ABCE1 (R2, K15, N17, E19, K59) and the phosphate backbone as well as 2’OH groups of rRNA (Figure 2A). Similarly, also the interaction sites between ABCE1 and uS12 are conserved (P25, R28, and S29 of ABCE1 to Q76 and H100 of uS12) (Figure 2A).”

• *In Fig. 3, A138 is not labeled in panels A and D despite being mentioned on p. 7, third paragraph.*

Reply: As requested, we now labeled A138 in Figure 3A and D.

• *Similarly, N316 and S580 are not labeled in Fig. 3A, despite being mentioned at the bottom of page 7.*

Reply: As shown in Figure 2, residue N316 is part of the interaction between hinge 1, the cantilever β -sheet, and rRNA h5. The interactions described at the bottom of page 7 were referring to residue N305 interacting with R572 and rRNA h14 as shown in Figure 3C. We corrected the main text in the revised manuscript accordingly.

Residue S580 is at the N-terminal end of helix 25; in the view presented in Figure 3A, the side chain of this residue is not visible and can thus not be labeled. However, it is visible in Figure 3D, where we labeled it now. We changed the revised text accordingly.

• *In Fig. 3H, it would useful to include density for the ligand since the text mentions that the high-resolution allows the identification of the bound nucleotide analogue and an Mg^{2+} ion.*

Reply: We agree with the referee that showing the density would be appropriate to support our claim. To that end, we added an additional Expanded View Figure (Figure EV5). This figure shows density and model for the NBSs in the same view as Figure 3H and an additional view, in which the densities for the isolated magnesium-nucleotides in the two NBSs are shown more clearly. Adding the density to Figure 3H itself, however - as suggested by the referee - is barely possible since this Figure shows an overlay of the two NBSs. Moreover, adding density to Figures 3I and 3J, which are showing the ATP coordination in both sites, is unfavorable, since these panels are too small to show the density appropriately. Thus, we found it more appropriate to add these important display items in a larger format in the context of an Expanded View Figure.

- *In Fig. 4J, G216 should be labeled*

Reply: We assume the referee is referring to Figure 3J. Here, we labeled G216 as requested and also the equivalent residue G463 in NBSI in Figure 3I.

- *In Fig. 5, shouldn't the arrows reflecting domain movements in ABCE1 during splitting go in the opposite direction? At least this is what I would think based on the accompanying movie.*

Reply: We apologize for the confusion. The black arrows were supposed to illustrate the allosteric communication pathway in ABCE1 from NBSII to NBSI and to the FeSD, and not the domain movements. But we agree with the reviewer that showing the domain movement is more intuitive. Therefore, we changed the arrows to reflect the NBD and FeSD movement accordingly.

2nd Editorial Decision

13th Jan 20

Thank you for submitting your revised manuscript for our consideration, it has now been seen once more by the original referees (see comments below). I am pleased to say that the referees find that their concerns have been satisfactorily addressed and now support publication. I would therefore like to ask you to now address some editorial issues that are listed in detail below. Please make any changes to the manuscript text in the attached document only using the "track changes" option. Once these minor issues are resolved in a final revised version, we will be happy to formally accept the manuscript for publication.

REFEREE REPORTS

Referee #1:

All the reviewer's comments were addressed sufficiently. Thus, I highly recommend publication of this work in EMBO Journal.

Referee #2:

The authors have successfully addressed all of the points I raised and I am fully satisfied with the revised version of the manuscript. Publication of this work will be of interest to the readership of EMBO Journal.

Corresponding Author Name: Robert Tampé

Manuscript Number: EMBOJ-2019-103788